# Strategies to Tackle Antimicrobial Resistance: The Example of *Escherichia coli* and *Pseudomonas aeruginosa*

**DOI:** 10.3390/ijms22094943

**Published:** 2021-05-06

**Authors:** Giada Antonelli, Luigia Cappelli, Paolo Cinelli, Rossella Cuffaro, Benedetta Manca, Sonia Nicchi, Serena Tondi, Giacomo Vezzani, Viola Viviani, Isabel Delany, Maria Scarselli, Francesca Schiavetti

**Affiliations:** 1GSK Vaccines, 53100 Siena, Italy; giada.x.antonelli@gsk.com (G.A.); luigia.x.cappelli@gsk.com (L.C.); paolo.x.cinelli@gsk.com (P.C.); rossella.x.cuffaro@gsk.com (R.C.); benedetta.x.manca@gsk.com (B.M.); sonia.8.nicchi@gsk.com (S.N.); serena.x.tondi@gsk.com (S.T.); giacomo.x.vezzani@gsk.com (G.V.); viola.x.viviani@gsk.com (V.V.); isabel.x.delany@gsk.com (I.D.); 2Department of Biotechnology, Chemistry and Pharmacy, University of Siena, 53100 Siena, Italy; 3Department of Pharmacy and Biotechnology (FABIT), University of Bologna, 40126 Bologna, Italy; 4Department of Life Sciences, University of Siena, 53100 Siena, Italy

**Keywords:** *Escherichia coli*, *Pseudomonas aeruginosa*, antimicrobial resistance (AMR), antigen identification, vaccine, monoclonal antibodies (mAbs), outer membrane vesicles (OMV)

## Abstract

Traditional antimicrobial treatments consist of drugs which target different essential functions in pathogens. Nevertheless, bacteria continue to evolve new mechanisms to evade this drug-mediated killing with surprising speed on the deployment of each new drug and antibiotic worldwide, a phenomenon called antimicrobial resistance (AMR). Nowadays, AMR represents a critical health threat, for which new medical interventions are urgently needed. By 2050, it is estimated that the leading cause of death will be through untreatable AMR pathogens. Although antibiotics remain a first-line treatment, non-antibiotic therapies such as prophylactic vaccines and therapeutic monoclonal antibodies (mAbs) are increasingly interesting alternatives to limit the spread of such antibiotic resistant microorganisms. For the discovery of new vaccines and mAbs, the search for effective antigens that are able to raise protective immune responses is a challenging undertaking. In this context, outer membrane vesicles (OMV) represent a promising approach, as they recapitulate the complete antigen repertoire that occurs on the surface of Gram-negative bacteria. In this review, we present *Escherichia coli* and *Pseudomonas aeruginosa* as specific examples of key AMR threats caused by Gram-negative bacteria and we discuss the current status of mAbs and vaccine approaches under development as well as how knowledge on OMV could benefit antigen discovery strategies.

## 1. Introduction

Throughout the evolution of human beings, infectious diseases have had a major determining effect on the age of our population [1]. Nowadays, an individual born in a high-income country can hope for a lifespan of more than 85 years thanks to improved hygiene, antibiotic and vaccination implementation. As a matter of fact, from a global health perspective, the discovery of antibiotics and the use of vaccines are considered the two more significant advances in medical care that have enabled the dramatic reduction in morbidity and mortality caused by infectious diseases [2]. Nevertheless, the effectiveness of antibiotics has been impaired as an increasing number of pathogenic bacteria are not killed by previously effective drugs, to the point that our lives can be seriously threatened. In fact, the emergence of antimicrobial resistance (AMR) is increasing the number of deaths and the spread of previously uncomplicated infectious diseases to treat [3,4]. Moreover, the risk of AMR during medical procedures such as surgeries, organ transplantations and immunosuppressive chemotherapies is becoming more significant and, in some cases, even prohibitive. Therefore, the evolution of AMR is becoming a dramatic threat to our lifespan, similar to the pre-antibiotic era [5]. Currently, AMR pathogens already determine globally 700,000 deaths/year, with 10 million deaths/year projected by 2050—a number even higher to that caused by cancer today [6]. In 1946, Alexander Fleming predicted the global AMR threat with the sentence “There is probably no chemotherapeutic drug to which in suitable circumstances the bacteria cannot react by, in some way, acquiring ‘fastness’ (resistance)” [7]. Nowadays, the time necessary for bacteria to become resistant to newly introduced antibiotics is getting shorter. In fact, the incorrect use or abuse of antibiotics is continuously imposing evolutionary pressure for the generation and transmission of resistant pathogens [8].

A plethora of mechanisms have led to antimicrobial resistance in microorganisms. The classification includes intrinsic, acquired and adaptive resistance mechanisms. The intrinsic antibiotic resistance refers to the innate ability of a bacterium to curtail the efficacy of a specific antibiotic through inherent structural or functional features. By contrast, the acquired resistance relies on the acquisition of new functions. When bacteria become resistant to one or more antibiotics that were initially effective, they are referred to as multi-drug resistant and often called “superbugs” [9]. This can occur through mutational changes or acquisition of resistance genes via horizontal gene transfer. These acquired genes are frequently localized within mobile genetic elements, such as plasmids, transposons, phages, insertion sequences, genomic islands and integrative and conjugative elements [10,11]. Horizontal gene transfer is not only frequent among microorganisms within the same genus, but also occurs among evolutionarily distantly-related bacteria. Bacteria can deal with the action of antibiotics by exploiting several mechanisms based on: (1) hydrolysis or structural modification of the drug leading to its inactivation, (2) lower membrane permeability or overexpression of efflux pumps preventing the access to the target, (3) mutation or post-translational modifications of the antibiotic targets [5,12,13]. Lastly, adaptive resistance is the ability of a bacterium to tackle antibiotics through transient alterations in gene expression in response to specific stimuli. The acquired phenotype is reversible and when the stimulus is removed the inherent bacterial sensitivity is restored [14]. The main mechanisms of this type of resistance are the formation of biofilm and the generation of persister cells. A biofilm is an aggregation of bacteria present on a living or non-living surface, that are encased within a self-produced matrix of extracellular polymeric substances, including proteins, exopolysaccharides, metabolites and extracellular DNA [15]. The microbial cells grown in biofilms are less sensitive to antimicrobial agents and host immune responses than the planktonic free-floating cells. This is due to the fact that biofilm protects bacteria by preventing antibiotic penetration, altering microenvironment to induce slow growth of bacteria, determining an adaptive stress response and differentiation of the bacterial cells toward the persister phenotype [16]. These bacterial cells are phenotypic variants tolerant to high concentrations of drugs even if not genetically resistant to them [17]. Due to the dormant state, such persisting cells are slow-growing, metabolically inactive and remain viable, thus repopulating biofilms [18]. Nevertheless, this phenotype is not limited to biofilm lifestyle, as a variety of other hostile environments may act as general activators of persister cells formation, including oxidative stress [17] and exposure to sub-lethal levels of antibiotics [19]. Regardless of the stimuli, the achieved tolerance of bactericidal drugs through dormancy is mainly due to bacterial toxin/antitoxin modules [18]. Therefore, the residual persister cells are considered the cause of the recalcitrance of chronic infections due to the failure of antibiotic treatments to completely eradicate pathogens from infected tissues [20].

Given the alarming rate in antibiotic resistance cases, in 2017 the World Health Organization (WHO) communicated a prioritization list of pathogens (classified as critical, high and medium priority) to guide the discovery of appropriate prophylactic and therapeutic strategies [21]. As part of the critical category of the WHO’s priority list, *Pseudomonas aeruginosa* and *Enterobacteriaceae* are becoming a healthcare concern due to the prominent level of resistance to many commercially available drugs [22,23]. They also represent two of the ESKAPE *(Enterococcus faecium*, *Staphylococcus aureus*, *Klebsiella pneumoniae, Acinetobacter baumannii*, *Pseudomonas aeruginosa* and *Enterobacter spp)* Gram-negative pathogens, so-called to emphasize their ability to evade the antibiotic activities leading to difficulties in treating such hospital infections [24]. Therefore, the development of new effective medical interventions as well as the discovery of new chemical compounds with an appropriate balance of antibacterial activity, drug metabolism, pharmacokinetics properties and safety it is a daunting task [21,25]. Even if a new successful drug is found, its clinical utility will decline as resistance inevitably arises [26].

Vaccines may become a valuable and effective weapon to fight AMR. Unlike antibiotics, vaccines are conceived to prevent diseases making antibiotic resistance mechanisms of less concern. Their prophylactic use enables the host to mount a specific immune response at the beginning of the infection, hence limiting the use of future antibiotic treatments potentially responsible to increase antimicrobial resistance threats [27]. An alternative strategy is based on the use of monoclonal antibodies (mAbs) as a preventative measure before some medical procedures such as invasive surgery, or as therapeutic medical intervention, used after the onset of an infection [28]. Monoclonal antibodies may act via the neutralization of key toxins or virulence factors or through inducing clearance of the bacteria activating the complement-mediated bacterial lysis or opsonophagocytosis. 

Here we discuss possible strategies to tackle antimicrobial resistance against two ESKAPE pathogens: extra intestinal pathogenic *Escherichia coli* and *Pseudomonas aeruginosa*, focusing on the state of art of mAbs and vaccines. The putative future impact that outer membrane vesicles (OMV) may have on both pathogens as tool for antigen discovery and as vaccine platform will also be discussed. OMV are indeed spherical particles derived from the outer membrane lipid bilayer of Gram-negative bacteria [29] (Figure 1). They generally contain all surface antigens both protein or polysaccharide that in nature are on the surface of the bacterium and as such can provide a useful tool for both antigen discovery and delivery [30]. During the last decades they have been exploited for a wide array of different biomedical applications spanning from vaccine and drug delivery vehicles to the recent use of cancer immunotherapy [31,32,33,34,35,36,37]. In particular, when isolated from the disease-causing pathogens, OMV can be used as a vaccine component per se by providing immunostimulatory agents and protective antigens [32,36]. Likewise, their versatility and “plug and play” features have made them attractive as platform for the display of heterologous and homologous bacterial, viral and even cancer antigens [33,34,35,36,37].

## 2. Epidemiology and Pathogenesis of *Pseudomonas aeruginosa* and *Escherichia coli*

### 2.1. Pseudomonas aeruginosa

*P. aeruginosa* is a Gram-negative bacterium capable of surviving in a wide range of environments [38]. In fact, this ubiquitous microorganism can be found in a wide variety of ecological niches such as plants, animals and humans, due to its metabolic versatility. This adaptability to environmental changes can be explained by its enhanced coding capability [39]. Indeed, the genome of *P. aeruginosa* (5.5–7 Mbp) is relatively large considering other sequenced bacteria such as *E. coli* (4.6 Mbp). *P. aeruginosa* rarely affects healthy individuals, but has been recognized as an opportunistic human pathogen that causes high morbidity and mortality in immunocompromised and hospitalised individuals [40]. This pathogen can cause both acute and chronic infections. During acute infections, it colonizes different anatomical sites, among others urinary tract, skin, eye, heart, ear, airway and lung tissues of immunocompromised individuals. Chronic infections are common in the lungs of patients with cystic fibrosis and bronchiectasis, and it accounts for 5% of cases of chronic obstructive pulmonary disease [41,42]. Interestingly, the type of infection is independent of the pathogen genotype, but possibly linked to the host health status and the lifestyle adopted by the bacteria when colonizing the host [43]. Acute infections are mainly associated with bacteria assuming a planktonic lifestyle, while biofilm plays a major role in persistent infections. Generally, in these two stages bacteria are characterized by different physiology and adapted behaviour. Planktonic bacteria are endowed with aggressive host-invasion strategies, while biofilm-forming bacterial cells are equipped with less cytotoxic and immune evasion strategies, usually observed in the recalcitrance of infections. Nevertheless, this model in which *P. aeruginosa* switches between these two lifestyles might be a simplified and static view of a more complex process, where acute and chronic virulence traits can co-exist [39].

Multidrug-resistant (MDR) or extensively drug-resistant (XDR) *P. aeruginosa* strains are increasingly prevalent in chronic and nosocomial infections such as wounds and burn patients and are associated with increased morbidity and mortality [44]. Despite the presence of geographical differences, the prevalence of MDR/XDR strains is increasing due to the highly frequent mutator phenotypes. These strains are characterized by enhanced rates of spontaneous mutations which result in resistance to many of the available medical options (e.g., carbapenemase- or extended-spectrum B-lactamases strains) [45]. Three major MDR/XDR high-risk clones are ST11, ST175 and ST235. The latter has the widest distribution, being found in all five continents [46]. Therefore, MDR/XDR global clones disseminated worldwide in hospital settings, have made previous uncomplicated infections untreatable.

### 2.2. Escherichia coli

*E. coli* is a Gram-negative bacterium commonly found in the gut of human and other warm-blooded animals. Most *E. coli* strains are harmless commensals, however, among them, some pathogenic variants can cause severe intestinal or extraintestinal infections and diseases [47]. Relatively to the extraintestinal pathogenic *E. coli* (ExPEC) strains, four main pathovars have been identified: neonatal meningitis *E. coli* (NMEC), uropathogenic *E. coli* (UPEC), septicemia-associated *E. coli* (SePEC) and avian pathogenic *E. coli* (APEC) [48]. These bacteria are the most frequent causes for the onset of severe diseases such as meningitis in new-borns, cystitis or acute pyelonephritis and sepsis [49]. EXPEC causes a high incidence of human infections globally [24,50,51,52] and, due to the acquisition of antibiotic resistance plasmids such as extended spectrum β-lactamase and mobilized colistin resistance, these bacteria have been recognised as an important AMR threat [24,53].

In the first years of this millennium, urinary tract infections (UTIs) affected 150 million people yearly worldwide, resulting in at least USD 6 billion dollars (about USD 18 per person) of direct medical expenditures in the US only [28,54]. Nowadays this number remains unvaried, underling the fact that UTIs are still a leading cause of morbidity in people of all ages, and pointing to the need of a specific treatment of this pathology [55,56]. UTIs can be divided into complicated and uncomplicated infections. The former is mostly due to indwelling catheters (called catheter associated UTI, CAUTI) and are the most common cause of secondary bloodstream infections (BSI) [57,58]. The latter are typical of otherwise healthy individuals and can be divided into lower UTIs (cystitis) and upper UTIs (pyelonephritis) [59,60]. Counting all forms of UTIs, four out of five are caused by uropathogenic *E. coli* (UPEC) infections [55,56]. A main source for UTIs is thought to be the gut microbiota *E. coli* [55,61]. While UTIs are commonly treated with the use of antibiotics, the appearance of multidrug-resistant UPEC strains, that are now increasing in many countries [62], represent a significant global risk. 

Moreover *E. coli* is the second leading pathogen causing neonatal meningitis. Several features, such as mortality (10%) and morbidity (30%) rates, serious neurological complications (mental retardation, hearing loss and cortical blindness) occurring in 30–58% of the survivals, as well as the increasing number of antibiotic resistant strains make this pathogen extremely threatening [47,63,64,65]. The treatment of choice for infections caused by NMEC remains the use of antibiotics. Among them, ampicillin and gentamicin are the broad-spectrum antibiotics currently used for empiric neonatal sepsis treatment. In 2016 a case of aggressive neonatal meningitis caused by a multidrug-resistant strain of *E. coli* was reported [66]. This event may be the tip of the iceberg of a more diffuse phenomenon of multidrug-resistance acquisition by NMEC, which would represent a major global health care danger.

## 3. State of the Art of Vaccines and mAbs to Tackle *Pseudomonas aeruginosa* and *Escherichia coli*

As previously discussed, AMR has now become a public health concern, especially since many bacterial strains show resistance to a broad spectrum of antibiotics. In most cases, it is the result of pathogens evolving resistance through horizontal gene transfer and adapting to drugs due to their abuse, often deviating from the indicated dosage and inappropriate prescribing in medical practice. One further cause is the widespread and uncontrolled use of drugs in animals for increased meat production [6,67,68,69,70,71,72]. In this scenario, vaccines and mAbs represent important immunotherapeutic tools.

The provision of an effective vaccine to protect populations has been recognized as an essential tool to fight AMR [73]. So far vaccines have proven more robust against evolution of resistance than drugs, generally providing sustained disease control. Two main reasons have been put forward for this [74]. Firstly, while antimicrobials tend to be administered when the pathogen population is already large, increasing the probability of evolution of drug resistance and transmission, the prophylactic nature of vaccines reduces the opportunities for resistance to emerge and spread. In addition, the vaccine preventive mechanism leads to a decreased need for antibiotic prescriptions and a decrement in selective drug pressure responsible for resistant strains [72,75,76]. Secondly, multivalent vaccines may induce immune responses against multiple targets on a pathogen, while drugs tend to target specific singular functions.

Moreover, for the preventive, pre-emptive or therapeutic treatment of infections caused by multi-resistant Gram-negative bacteria the development of mAbs is also a promising alternative to antimicrobials. Due to significant advancement in technologies that overcome high production costs and limited efficacy, the limited number of mAbs to infectious diseases currently available is about to change. Monoclonal antibodies target specific surface antigens not usually recognized by antimicrobials and are therefore effective against bacteria which have acquired broad range resistance and can hence be used as a last resort [77]. Furthermore, the use of mAbs recognizing specific targets can prevent the impact on the human flora, unlike antibiotics. Their immediate protective effect once administered means that they can be used in situations of planned or emergency medical interventions such as surgery, ventilator or catheter use, where nosocomial infections with AMR bacteria present a huge risk. In conclusion, their therapeutic use could circumvent AMR in bacteria and greatly reduce the prescription of antibiotics, thus possibly leading to a reduction in resistant strains. On the other hand, mAbs present some limitations due to: high specificity towards the target pathogen (making it necessary to identify the infecting pathogen before any therapy is applied), type of administration (mostly intravenous, which is not ideal for all types of patients) and their efficacy, particularly against more complex bacteria, where targeting one corresponding single epitope, especially in terms of its conservation and expression, may not be enough and multivalent formulations may be needed [77,78]. 

The discovery and development of vaccines or monoclonals against AMR pathogens remain a challenging and a time-consuming goal, but it is becoming increasingly urgent due to the emergence of untreatable infections. So far, we do not have effective licensed vaccines or monoclonals to the *E. coli* or *P. aeruginosa* AMR threats. However, there are and have been many candidates in preclinical and clinical trials and we discuss these below. 

### 3.1. Pseudomonas aeuroginosa Vaccines and mAbs

The use of vaccines against *P. aeruginosa* would overcome many of the problems associated with antibiotic resistance by eliminating or greatly reducing the need to use antibiotic agents and by providing effective protection against multidrug-resistance infections. *P. aeruginosa* strains are classified into twenty serotypes based on the type of O-antigen and each of them has several subtype strains with subtle variations, leading to more than thirty subtypes. Over the years it has been demonstrated that the O-antigen is able to mediate a high level of antibody immunity during *P. aeruginosa* infections. The first O-antigen based vaccines prepared from lipopolysaccharide (LPS) (e.g., Pseudogen heptavalent vaccine) showed some efficacy, but their toxicity was the major limitation [79,80,81,82]. Moreover, one of the main issues associated with O-antigen-based vaccines is the limited coverage determined by the polysaccharide variability. For this reason, it has been estimated that at least ten major O-antigen variants should be theoretically combined to protect against the most common serotypes [83]. However, it has also been observed that the combination of multiple purified O-antigens from strains within the same sub-variants tends to diminish the mouse immune response to each individual component [84]. To date several *P. aeruginosa* vaccines have been tested in cystic fibrosis patients with little success [85]. Cryz and co-workers evaluated whether LPS-specific antibodies elicited by an octavalent O-antigen-exotoxin A conjugate vaccine (Aerugen) could prevent colonization of *P. aeruginosa* in children with cystic fibrosis. Although the studies were initially promising, subsequent results did not show a delay in colonization by *P. aeruginosa* [86]. Of note, among the vaccines tested so far, the bivalent flagellum vaccine (IMMUNO) tried on a large randomized trial, while not meeting primary endpoints, showed a small but significant reduction in *P. aeruginosa* infection only against strains carrying the same flagella type [87].This suggests that the flagella, which show less diversity with respect to the O-Antigen, could be considered for a multivalent formulation covering all variants. However, up to now, no other bivalent or multivalent flagellar-based vaccines have been produced [88]. The recombinant IC43 vaccine for the treatment of *P. aeruginosa* infections of mechanically ventilated Intensive Care Unit (ICU) patients consists of a fusion protein vaccine composed by OprF and OprI proteins. To assess the efficacy, immunogenicity and safety of the recombinant IC43 vaccine, a randomized, multicenter, placebo-controlled, double-blind, phase 2/3 trial was conducted. IC43 demonstrated a significant immunogenic effect in ventilated ICU patients without safety concerns. In addition, mortality was reduced compared to placebo (more pronounced in patients with infections), although there were no significant differences in the rates of *P. aeruginosa* infection [89,90,91,92]. The OprF/I vaccine induced the production of opsonic antibodies as well as antibodies that inhibited IFN-γ binding and thus interfered with a virulence mechanism as well as a T-cell response. Thereby, based on safety and immunogenicity results obtained so far, the IC43 vaccine appeared promising. Although, even if for many decades, efforts have been made to discover an effective vaccine strategy to protect patient populations at risk of infection with *P. aeruginosa*, presently there are no licensed vaccines (Figure 2).

Over the years, several monoclonal antibodies to treat *P. aeruginosa* infections have been tested in clinical trials [93]. MEDI3902 is a bivalent monoclonal antibody developed by MedImmune LLC. It recognizes and binds the two virulence factors Psl and PcrV. The Psl exopolysaccharide plays a key role in both pathogen colonization and adhesion to host tissues and it is also involved in immune evasion and biofilm formation. The PcrV protein plays a vital role in host cell cytotoxicity and belongs to the type III secretion system. Its presence has been correlated with increased disease severity following *P. aeruginosa* infection [78,94,95]. Preclinical studies have shown that MEDI3902 is able to induce protection against lethal *P. aeruginosa* infection in the mouse model of pneumonia by maintaining lung integrity, preventing spread and reducing the bacterial load [78,95]. Initial clinical studies in healthy adults showed no serious treatment-emergent adverse events and an increase in anti-cytotoxic and opsonophagocytic activity was observed in a dose-dependent manner following MEDI3902 administration [95]. Phase 2 study was carried out in intensive care subjects undergoing mechanical ventilation where reduction of pneumonia incidence caused by *P. aeruginosa* was evaluated. The conclusion of the study was that a single dose of intravenously administered MEDI3902 did not achieve the primary endpoint, however, it suggested that MEDI3902 may be effective in ICU patients who have a lower level of baseline inflammation [96].

Despite all the efforts none of the mAbs has been licensed so far due to suboptimal levels of protection observed or limited coverage (Figure 2). As such the development of highly effective mAbs against *P. aeruginosa* remains a challenging goal. Likewise, it outlines the need to discover new antigen targets to overcome their high variability and low expression in vivo that likely limited so far the clinical applicability of mAbs to pseudomonal infection.

### 3.2. Escherichia coli Vaccines and mAbs

Due to the complexity and heterogeneity of ExPEC, the development of vaccines to overcome the multidrug resistance threat has been unsuccessful so far. Different efforts have concentrated around vaccines to develop immunity against the O-antigens of different ExPEC serotypes, initially bivalent conjugate formulations and later with increasing valency [97,98] (Figure 3). Different vaccines are under development or available to prevent recurrent UTIs that base their action on surface antigens or whole inactivated bacteria [99]. Uro-Vaxom is an oral vaccine, licensed in over 30 countries, which is composed of bacterial crude extracts from eighteen UPEC strains and has been shown in several randomized placebo controlled trials to reduce the frequency of UTI recurrence [100,101,102,103,104,105]. Urovac is a vaginal mucosal multivalent vaccine made up of whole inactivated bacteria including six strains of heat inactivated UPEC along with four other strains of uropathogenic bacteria [106,107]. Data reported on three clinical trials demonstrated a reduction of recurrent UTIs during a 6-month period after immunization with Urovac [108]. MV140, also known as Uromune, is a sublingual vaccine composed of a mixture of inactivated *Escherichia coli*, *Klebsiella pneumoniae*, *Proteus vulgaris* and *Enterococcus faecalis* currently pre-licensed in phase 3 stage. Although randomized placebo-controlled clinical trials are still underway, preliminary data from retrospective and prospective uncontrolled studies reported a significant efficacy of the vaccine to prevent recurrent UTIs in women between 16 and 97 years old [109,110]. However, the compliance of this treatment could be questioned considering the 3-month daily administration period. ExPEC4V is the only UTI vaccine under retrospective and prospective uncontrolled studies that exploits the bioconjugation technique. O-antigens from four *E. coli* serotypes (O1A, O2, O6A and O25B) which cover around 30–35% of UTIs caused by *E. coli* were conjugated in vivo with *P. aeruginosa* exoprotein A as a carrier [111]. Indeed, besides its use as an antigen per se in pseudomonal vaccine [86], in the context of ExPEC4V, T-cell epitopes present in the exoprotein A have been exploited to enhance the T cell response toward the bioconjugate LPS [111]. To date, three clinical trial studies were performed to assess the safety, tolerability and immunogenicity of ExPEC4V [112,113,114]. Data showed that the vaccine is well tolerated without significant adverse effects and that it can elicit a high immunogenic response across all four serotypes with a durability of one year, but it failed to show a reduction in UTI recurrence rates. Finally, a vaccine based on FimCH, a pili protein of *E. coli* responsible for adhesion and colonisation of the bladder, is also under development against UTIs caused by *E. coli*. After successful phase 1 trials where it was seen to be safe and immunogenic inducing functional antibodies it is moving to phase 2 [115].

Since *E. coli* is one of the principal commensals in human, the use of monoclonal antibodies, which have a narrow spectrum in comparison to classical antimicrobial drugs, could be preferred (Figure 3). Moreover mAbs offer advantages in terms of limitation of selective pressure and precise delivery through engineering [116]. Nonetheless, there is only one mAb (A1124) currently proposed to treat multidrug resistant ExPEC sequence type 131-H30 [117], whose activity was confirmed against a colistin-resistant strain [118]. Its specific action on the LPS O-antigen (O25b) of *E. coli* points out the importance of saccharide epitopes to elicit protective mAbs [119]. Finally, preclinical studies have shown that a promising anti-FimH mAb was able to prevent the invasion of UPEC strains in mice, inhibiting the pathogenic activity of the bacteria [120].

Interestingly, another universal antigen that could target several AMR pathogens is the surface polymer poly-β-1,6-*N*-acetylglucosamine (PNAG), a polysaccharide expressed on the surface by many AMR bacteria and a component of biofilms made by numerous Gram-negative bacterial species [121]. This represents one of the first cross-species protective antigen to be investigated. Vaccine approaches with PNAG were initially tested in farm animals [122]. Furthermore, a vaccine with PNAG as principal immunogen (AV-0328) recently passed phase 1 and 2 in a clinical study (NCT02853617). A human IgG1 mAb (F598), generated against a deacylated form of this antigen, had a protective effect in mice infected with different AMR bacteria [123]. Therefore, F598 successfully passed phase 1 [122] and is pursued in phase 2 clinical trials as passive protection studies [124].

## 4. Outer Membrane Vesicles as Vaccine Platform

In the past years, outer membrane vesicles (OMV) have represented an attractive and cost-effective approach for vaccine development due to their built-in adjuvanticity, immunogenic properties and ability to induce humoral and cellular immune responses [32,36]. Moreover, the simultaneous presence and delivery of multiple antigens on OMV reduce the possibilities for the pathogen to generate vaccine escape mutants [29]. OMV first clinical use was to tackle *Neisseria meningitidis* outbreaks and from them three OMV-based vaccines (MenBvac, MeNZB and VA-MENGOCOC-BC) have been deployed to successfully fight meningococcal outbreaks [125,126,127]. Given their success OMV have been included as one of the components of the licensed broad-spectrum meningococcal B vaccine Bexsero, currently on the market [128,129]. Since then, OMV-based vaccines have received renewed attention and nowadays more recent approaches involve the generation of genetically engineered strains where the endotoxicity of the LPS is attenuated. As alternative, the use of synthetic and semi-synthetic liposomes has been proposed to remove LPS and related toxicity issues [130]; however currently there is a good understanding of how to genetically modify LPS biosynthesis in OMV strains to reduce its endotoxicity and maintain adjuvanticity [131]. Nevertheless, the use of bacteria-derived OMV offers the opportunity of modulate antigen content leading to overexpression of protective targets and removal of unwanted antigens [36]. With regard to this, genetically engineered OMV-based strategies against the shigella and meningococcus Gram-negative pathogens are currently in the clinical phase [32]. 

Several OMV candidate vaccines have been shown preclinically to be protective against the Gram-negative pathogens [32]. *Pseudomonas* OMV induce protective responses to lung infection in a mouse challenge model [132] and *E. coli* OMV have been used as a platform for delivery of heterologous proteins and polysaccharide antigens [133]. Nevertheless, the high protein identity shared between *E. coli* species has raised concerns that *E. coli* OMV may induce an immune response against the commensal population [134,135]. Recently synthetic biology has been used to remove up to fifty-nine dispensable endogenous *E. coli* proteins that could potentially enhance the antigen specific immune response and at the same time they may theoretically reduce the cross recognition with natural flora [136]. Although, the manufacturing downstream process required for OMV production is potentially cheap and relatively unsophisticated, current bottlenecks to develop OMV vaccine candidate for *Pseudomonas* at the preclinical level, are the yields and the detoxification strategies. In fact, low yields are generally obtained from *Pseudomonas* and an overblebbing genetic strategy, similar to that of *E. coli* for which extensive knowledge is available in literature [137,138,139,140,141], remains unidentified. Moreover, the purification of *P. aeruginosa* OMV represent a serious challenge [142,143] due to its tendency to secrete large amounts of extracellular factors such as enzymes, toxins and exopolysaccharide, obviously impairing OMV purification.

Recently OMV are receiving more and more attention not only as a platform for delivery of multivalent surface antigens from the target pathogenic bacteria, but also as tools in the search of effective candidates. As OMV resemble the composition of bacterial outer membrane, displaying a complex array of surface antigens in their native conformation/orientation, proteomic characterization of *P. aeruginosa* and *E. coli* OMV could be a powerful strategy to identify pathogen-specific outer membrane proteins (OMPs) [32,144,145]. On one hand, pseudomonal OMV, released both in planktonic and sessile phases, may recapitulate the complete antigen repertoires occurring in acute or chronic infections [146]. On the other hand, the use of human urine as culture medium for UPEC-OMV allow to identify specific OMPs expressed in a condition closer to the natural one occurring during the infections [134,145]. In addition, systematic analysis of UPEC vesicles from representative strains, may potentially cover the phenotypic diversity of all clinical isolates affecting the urogenital tract. Taken together, experimental settings that mimic in vivo environment coupled with in depth studies of OMV proteome may be instrumental to understand the mechanisms applied by bacteria to respond to environmental changes and to identify functional targets for drug and therapeutics development. In fact over the years, OMV have already confirmed their essential role in the identification of protective antigens against a broad range of pathogens [147,148,149]. 

In conclusion, OMV could exert an effective tool for both prophylactic vaccines and antigen identification for the development of therapeutic mAbs. Therefore, advancement in the engineering strategies for optimal OMV production, mAbs and vaccine development will be crucial in the coming years to address the very critical challenge of antimicrobial resistance.

## 5. Gram-Negative AMR Future Prospective

Since the discovery of penicillin by Fleming in 1929, many antibiotic molecules have been developed which have had a huge influence on human health throughout the world and an important economic impact in terms of costs of treatment and long-term hospital stay [13]. This is particularly true for infections caused by multidrug resistant Gram-negative bacteria, highlighted by *E. coli* and *P. aeruginosa* here, for which few alternative options are available [150]. Nosocomial infections caused by these bacteria are a demanding issue for our health, due to high number of resistance virulence traits [151]. The outer membrane of Gram-negative bacteria is the first barrier conferring resistance to a variety of drugs. Any alteration into their outer membrane, such as changing the hydrophobic properties or mutations in porins, efflux pumps and other antibiotic targets, may determine resistance [12,152]. Gram-positive bacteria which lack this important outer layer are more susceptible to antibiotics than Gram-negative ones. Among Gram-negative bacteria, *Enterobacteriaceae* and *P. aeruginosa* are of particular medical relevance causing, among others, ventilator-associated pneumonia, catheter-related bloodstream infections and other ICU-acquired sepsis such as urinary tract infections. These two bacteria have been recognized in the critical category of the WHO’s priority list and they also represent two of the ESKAPE pathogens, acronym that highlights the ability of these microorganisms to “escape” the activity of antibiotic molecules [21,22,23,24]. Genetic flexibility and stochastic variations of bacterial properties are the key for their success as human pathogens enabling them to adapt to any sites of infection [20,39,47]. In addition, their relatively large genome encodes a high number of virulence factors which make the development of effective medical strategies even more complex and challenging. To date, no effective medical interventions, neither therapeutic nor prophylactic, are available. Therefore, an in-depth understanding of the molecular mechanisms of ExPEC and *P. aeruginosa* pathogenesis along with their relevant virulence traits exploited in different stages of their infections, are of crucial importance. The development of in vitro, ex vivo and animal models similar to the colonized human niches, that mimic the specific infections caused in the target populations, should be improved. Due to a plethora of mechanisms of pathogenesis, formulations consisting of a combination of multiple effectors may be beneficial before moving to clinical studies in specific target populations and OMV represent an attractive multivalent platform. To date, few vaccines and mAb approaches are in development in clinical trial for *P. aeruginosa*. Most approaches are based on antigens that are exposed on *P. aeruginosa* or *E. coli* outer membrane, suggesting the feasibility to present them on the OMV surface. Regarding *E. coli*, all vaccines that are currently in clinical trials are composed by recombinant proteins, O-antigen conjugates or heat-inactivated bacteria. The only two mAbs which succeeded in phase 1 clinical trials target polysaccharides, suggesting their importance as virulence factors. However, due to the large heterogeneity of O-antigens relevant to pathogenic strains of both *E. coli* and *P. aeruginosa*, effective vaccines and mAb formulations should be highly multivalent to ensure adequate coverage. As LPS or capsular polysaccharides seem to play an important role in immunogenicity and given the possibility to engineer the OMV with polysaccharides as described by Feldman and colleagues [133], the implementation of OMV technology with glycoengineering to better resemble antigens presented on bacteria surface, could also be an attractive strategy.

In conclusion, the combination of new technologies for target identification, vaccine development and monoclonal antibodies may provide affordable solutions to tackle antimicrobial resistance representing a hope for the future. 

## Figures and Tables

**Figure 1 ijms-22-04943-f001:**
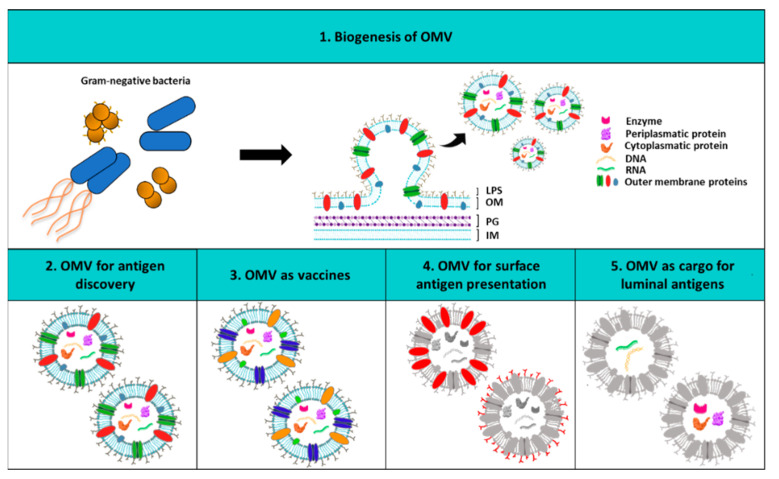
Outer membrane vesicles (OMV) biogenesis and biomedical applications. (1) The upper panel shows the structure of the OMV originating from the outer membrane of Gram-negative bacteria. In the other panels the wide range of OMV applications is depicted. In particular, (2) OMV can be used as an antigen discovery tool; (3) bacterial OMV are excellent vaccines since they trigger both humoral and cellular immune responses following the immunization; (4) OMV can be decorated on their surface with desired heterologous and homologous protein or saccharide antigens; (5) OMV can also function as cargo delivery of specific luminal recombinant antigens. LPS: lipopolysaccharide; OM: outer membrane; PG: peptidoglycan; IM: inner membrane.

**Figure 2 ijms-22-04943-f002:**
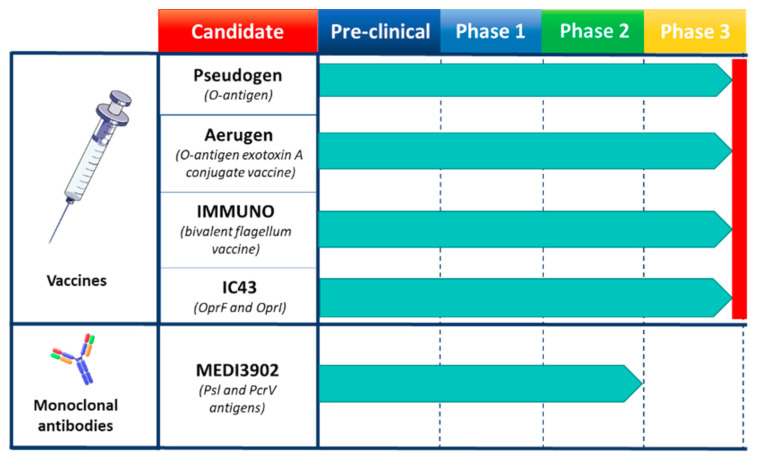
Human vaccines and monoclonal antibodies (mAbs) used to tackle *P. aeruginosa* infections. In the candidate sections are reported the trademarks and the specific targeted antigens. Pre-clinical and each clinical phase are represented. Red line indicates the phase at which studies were interrupted.

**Figure 3 ijms-22-04943-f003:**
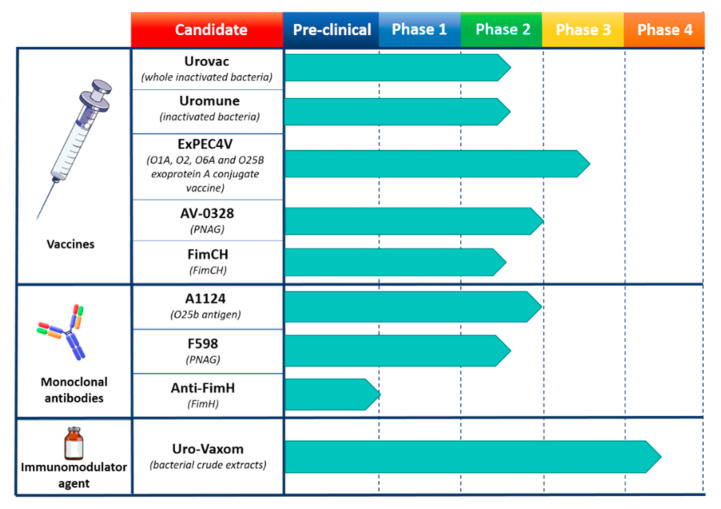
Human vaccines and monoclonal antibodies (mAbs) used to fight *E. coli* infections. The candidate section describes the trademarks and the specific targeted antigens. Pre-clinical as well as each clinical phase are represented. Arrows reaching dotted line indicate a completed study for that phase, conversely, studies are ongoing.

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
