# Peer review of "Strategies to Tackle Antimicrobial Resistance: The Example of Escherichia coli and Pseudomonas aeruginosa"

_ijms, 2021, doi:10.3390/ijms22094943_

Round 1
Reviewer 1 Report
Summary
The authors discuss general problems of antimicrobial resistance in bacteria and then explore some alternative therapeutic ideas for E. coli and Pseudomonas. They summarize some clinical trials involving vaccines and monoclonal antibodies for both pathogens, along with successes and setbacks. They then propose outer membrane vesicles (OMVs) as another possible alternative that has received little attention. The article nicely summarizes the alternative therapeutic strategies for both pathogens.
Major suggestions
- The article would benefit more readers if they had more precisely defined outer membrane vesicles in the introduction, cited relevant review articles on structure and function (with attention towards OMVs as antigen-delivery vehicles, delivery of antibiotic resistance proteins, and potential for DNA delivery in the center of the OMVs. The article would benefit also from a figure that captures OMVs from a structure-function point of view with suggested ideas for therapeutic potential (LPS or protein antigens on surface of OMVs, lumen OMV proteins, DNA in lumen of OMV that could encode an antigenic target, etc). After the abstract, OMVs are not mentioned again until line 120 and it is never defined and no mention is made to the counterpoint (membrane vesicles) in Gram positives.
- Elaborate more in the OMV section as to whether synthetic or semi-synthetic OMVs or specific genetic strains with well-defined OMVs might be possible that minimize potential for LPS side effects (since OMVs are likely to contain abundant LPS, since OMVs are derived from the outer membrane), minimize the disruption of E. coli commensals (carrying similar antigens), and other potential side effects.
- Fig. 1. Legend states “red line indicates the phase at which studies where interrupted, but in the online version of the manuscript I reviewed, there was no color in the figures. Please make sure that the area to be highlighted is clear, maybe replace red line with an X or something clearer.
- Lines 333-334. Elaborate just a little further on why Pseudomonas ExoA has features that make it so suitable for inclusion into vaccines for both Pseudomonas (Aerugen) and E. coli (ExPEC4V).
Minor suggestions
- Lines 166, 184, 191. “Meningitidis” should be meningitis
- Line 354. Remove 2nd “the” in sentence
Author Response
Dear Reviewer,
we warmly thank you for your comments that highly improved the quality of our review. Below you can find the point-by-point replies to the requests. We trust that now the manuscript reached the quality for publication on International Journal of Molecular Sciences.
Kind regards.
Major suggestions
- The article would benefit more readers if they had more precisely defined outer membrane vesicles in the introduction, cited relevant review articles on structure and function (with attention towards OMVs as antigen-delivery vehicles, delivery of antibiotic resistance proteins, and potential for DNA delivery in the centre of the OMVs. The article would benefit also from a figure that captures OMVs from a structure-function point of view with suggested ideas for therapeutic potential (LPS or protein antigens on surface of OMVs, lumen OMV proteins, DNA in lumen of OMV that could encode an antigenic target, etc). After the abstract, OMVs are not mentioned again until line 120 and it is never defined, and no mention is made to the counterpoint (membrane vesicles) in Gram positives.
In accordance with the suggestions, we added in the introduction, specifically in lines 122-132, more relevant information about OMV, focusing more on vesicles of Gram-negative bacteria.
“OMV are indeed spherical particles derived from the outer membrane lipid bilayer of Gram-negative bacteria [27] (Figure 1). They generally contain all surface antigens both protein or polysaccharide that in nature are on the surface of the bacterium and as such can provide a useful tool for both antigen discovery and delivery [28]. During the last decades they have been exploited for a wide array of different biomedical applications spanning from vaccine and drug delivery vehicles to the recent use of cancer immunotherapy [29-35]. In particular, when isolated from the disease-causing pathogens, OMV can be used as a vaccine component per se by providing immunostimulatory agents and protective antigens [30, 34]. Likewise, their versatility and “plug and play” features have made them attractive as platform for the display of heterologous and homologous bacterial, viral and even cancer antigens”.
As you kindly suggested, we added also a figure (now new Figure 1) on OMV biogenesis and potential applications.
- Elaborate more in the OMV section as to whether synthetic or semi-synthetic OMVs or specific genetic strains with well-defined OMVs might be possible that minimize potential for LPS side effects (since OMVs are likely to contain abundant LPS, since OMVs are derived from the outer membrane), minimize the disruption of E. coli commensals (carrying similar antigens), and other potential side effects.
In accordance with the suggestion, we added lines 407-413 to discuss the use of synthetic or semi-synthetic liposomes to reduce toxicity concerns, underling their different features compared to the bacterial OMV counterpart.
“As alternative, the use of synthetic and semi-synthetic liposomes has been proposed to remove LPS and related toxicity issues [129]; however currently there is a good understanding of how to genetically modify LPS biosynthesis in OMV strains to reduce its endotoxicity and maintain adjuvanticity [130]. Nevertheless, the use of bacteria-derived OMV offers the opportunity of modulate antigen content leading to overexpression of protective targets and removal of unwanted antigens [34].”
Moreover, we mentioned in line 422-425 the possible OMV genetic approaches put in place to reduce side effects on commensal species.
“Recently synthetic biology has been used to remove up to fifty-nine dispensable endogenous E. coli proteins that could potentially enhance the antigen specific immune response and at the same time they may theoretically reduce the cross recognition with natural flora [135].”
- 1. Legend states “red line indicates the phase at which studies where interrupted, but in the online version of the manuscript I reviewed, there was no color in the figures. Please make sure that the area to be highlighted is clear, maybe replace red line with an X or something clearer.
à As suggested, we redraw the Figure (now Figure 2) improving the thickness of the red bars.
- Lines 333-334. Elaborate just a little further on why Pseudomonas ExoA has features that make it so suitable for inclusion into vaccines for both Pseudomonas (Aerugen) and E. coli (ExPEC4V).
à We added in line 359-361 the reason behind the use of Pseudomonas Exoprotein A as a vaccine antigen both in Pseudomonas and in E. coli, providing the relative reference.
“Indeed, besides its use as an antigen per se in pseudomonal vaccine [84], in the context of ExPEC4V, T-cell epitopes present in the exoprotein A have been exploited to enhance the T cell response toward the bioconjugate LPS [109]”.
Minor suggestions
- Lines 166, 184, 191. “Meningitidis” should be meningitis à done
- Line 354. Remove 2nd “the” in sentence à done

Reviewer 2 Report
This manuscript is a well-written and well-organized review outlined antimicrobial resistance concerns, potential vaccines and non-antibiotic therapies to combat P. aeruginosa and E. coli infections. The article details potential vaccines of various forms and other strategies including the development of monoclonal antibodies as a preventative and treatment option. Only minor revisions are requested by this reviewer
Specific comments:
line 15: anti-microbial should be antimicrobial
line 60/61: replace nominated with "referred to as"
line 87: others should be other
line 88-89: Regardless the stimuli should be" Regardless of the stimuli"
line 100: replace GRAM with Gram
line 120: Define OMVs as this is the first use
lines 166 and 184: replace meningitidis with meningitis
line 205: replaced proved with proven
lines 252-254: please provide reference for this estimate
Author Response
Dear Reviewer,
We warmly thank you for your comments that highly improved the quality of our review. Below you can find the point-by-point replies to the requests. We trust that now the manuscript reached the quality for publication on International Journal of Molecular Sciences.
Kind regards.
Specific comments:
- line 15: anti-microbial should be antimicrobial -->done
- line 60/61: replace nominated with "referred to as"-->done
- line 87: others should be other --> done
- line 88-89: Regardless the stimuli should be" Regardless of the stimuli" --> done
- line 100: replace GRAM with Gram --> done
- line 120: Define OMVs as this is the first use --> done
- lines 166 and 184: replace meningitidis with meningitis --> done
- line 205: replaced proved with proven --> done
- lines 252-254: please provide reference for this estimate --> we have provided the reference 81 in the line 277-278.
“For this reason, it has been estimated that at least ten major O-antigen variants should be theoretically combined to protect against the most common serotypes [81]”.
